

# Multi-hazard fragility analysis for fluvial earthen dikes in earthquake and flood prone areas

Sergey Tyagunov[1], Sergiy Vorogushyn[2], Cristina Muñoz Jimenez[1], Stefano Parolai[1], Kevin Fleming[1]

[1] GFZ German Research Centre for Geosciences, Centre for Early Warning, Helmholtzstrasse 7, D-14467, Potsdam, Germany.

[2] GFZ German Research Centre for Geosciences, Section 5.4 - Hydrology, Telegrafenberg, D-14473, Potsdam, Germany.

*Correspondence to:* Sergey Tyagunov (sergey.tyagunov@gmail.com)

**Abstract.** The paper presents a methodological framework for multi-hazard fragility analyses for fluvial earthen dikes in earthquake and flood prone areas. The methodology and results are an integral part of the multi-hazard (earthquake-flood) risk study implemented within the framework of the EU FP7 project MATRIX (New Multi-Hazard and Multi-Risk Assessment Methods for Europe) for the area around Cologne, Germany. The study area covers the Rhine River reach and adjacent floodplains between the gauges Andernach and Düsseldorf. Along this domain, the inhabited areas are partly protected by earthen embankments (dikes or levees), which may be prone to failure in case of exceptional floods and/or earthquakes. The main focus of the study is to consider the damage potential of the dikes within the context of the possible interaction between the two hazards. The fragility of the earthen dikes is analyzed in terms of liquefaction potential characterized by the factor of safety. Uncertainties in the geometrical and geotechnical dike parameters are considered by using a Monte Carlo approach. The damage potential of the earthen structures is presented in the form of a fragility surface showing the damage probability as a function of both seismic ground shaking and flood water level. The obtained results can be used for multi-hazard risk assessment in earthquake and flood prone areas and, in particular, are intended for comprehensive risk assessment in the area around the city of Cologne.

## 1.    Introduction

The problem of risk assessment in areas affected by several natural perils can be discussed in two possible ways: on the one hand, one can consider the different types of hazards and risks independently, while on the other, the possible interactions between the hazards may be taken into account. The former approach is based on traditional methods of single-type hazard and risk assessment and presently is most commonly used by researchers and practitioners worldwide. The latter is used much more rarely, perhaps because it involves scenarios with obviously lower occurrence probabilities, which might, therefore, be underrated and sometimes unreasonably neglected. At the same time, the tragic lessons of past disasters show that in multi-hazard prone areas the dramatic consequences from single hazardous events may enormously increase due to



possible interactions between different types of hazards as well as initiation of cascading effects. Therefore, the potential interactions between hazards in multi-hazard prone areas should not be ignored in decision making.

A steadily increasing number of studies devoted to different aspects of multi-hazard risk assessment emphasize the raising awareness and interest of scientists and practitioners to quantifying multi-hazard scenarios, their effects and occurrence
probabilities. The earlier multi-hazard studies were solely based on the comparison of single-type hazard and risk assessments without considering interactions and potential cascading effects (e.g., HAZUS-MH, 2003, KATARISK, 2003, Grünthal et al., 2006). In the recent years, frameworks for the assessment of the interactions of multiple hazards have been developed for comprehensive risk assessment (e.g., Marzocchi et al., 2012, Selva, 2013, Mignan et al., 2014).

This research work was implemented in the frame of the EU FP7 project MATRIX (New Multi-Hazard and Multi-Risk
Assessment Methods for Europe, matrix.gpi.kit.edu). The present study, which is a part of the MATRIX Work Package 3 (Garcia-Aristizabal and Marzocchi, 2013), focuses on the problem of multi-hazard fragility analysis of fluvial earthen dikes (levees) representing the flood protection system in the area around the City of Cologne, Germany. We assess the fragility of those structures by taking into account potential flood and earthquake impacts.

The areas around Cologne as well as the city itself are regularly affected by flooding from the Rhine River (e.g., Fink et al.,
1996). Therefore, within the framework of the regional flood risk management program, the urban areas along the river are partly protected by a system of earthen dikes (or levees). The localities not protected by earthen embankments are typically protected either by concrete walls or by mobile flood protection walls; others are located on elevated banks and therefore they are not subject to considerable inundation.

Besides flood hazard, the areas around Cologne are exposed to other kinds of natural hazards; among them windstorms (e.g.,
Hofherr and Kunz, 2010) and earthquakes (e.g., Grünthal et al., 2009). Strong seismic events in the area are less frequent than floods or windstorms; however earthquakes (due to a higher damaging potential) may cause considerable losses in the area. Such conclusion is made from a comparison of the single hazards and risks in the area (Grünthal et al., 2006).

In the MATRIX project, for the first time to the author's knowledge the possible effects of spatio-temporal-coincidence of flood and earthquake hazards in the area are considered. The problem of temporal and spatial interaction between
earthquakes and floods is dual: on the one hand, one should consider how the existing earthquake hazard may influence the level of flood risk and on the other, how the presence of flood hazard may change the level of seismic risk. Generally, earthquake and floods can aggravate their impacts in a number of ways, for instance, through damage to the flood protection infrastructure by the earthquakes, through the increasing severity of local effects of ground shaking by the presence of inundated areas or by influencing the vulnerability (fragility) of the built environment. In the present work, the interaction of
the two hazards is accounted for only in terms of potential joint effect of seismic ground shaking and flooding on the flood protection system of earthen dikes.

The designated purpose of fluvial dikes (representing earthen embankments running along the river banks) is to confine the water flow up to a certain level and protect the built-up or arable areas against flooding. The physical (and functional) reliability of the dikes plays an important role in the reliable performance of the whole flood protection system. Therefore, a





comprehensive risk assessment should include the reliability analysis of the dikes, taking into consideration all the processes leading to possible breaching.

Dike may fail due to various damage mechanisms induced either by high water levels, moisture content in the dike core, intense rainfall influence  and/or earthquake impact (e.g., Armbruster-Veneti, 1999, Foster et al., 2000, Allsop et al., 2007, Briaud et al., 2008, Wolff, 2008, Van Baars and Van Kempen, 2009, Nagy, 2012, Huang et al., 2014). Correspondingly, a variety of approaches to account for different damage (failure) modes were developed and used for performance and reliability analyses of earthen structures under different hazard conditions. As has been already indicated, the focus of our research interest lies on earthquake and flood hazards.

Regarding the earthquake hazard considerations, one can refer to the review article of Marcuson et al. (2007), which traces the development of the state of practice in seismic design and analysis of embankment dams, starting from the fundamental publications of Newmark (1965) and Seed and Idriss (1971). Another overview of different approaches to seismic safety assessment of earthen embankments and dams can be found in Ozkan (1998), where the liquefaction phenomenon is indicated as the most important cause of the damage occurrence in earthquake prone areas.

Sasaki et al. (2004) described empirical and analytical methods used in Japan for estimating the settlement of river dikes due to liquefaction, considering both the probable subsidence of the bottom boundary and deformation of the dikes. With respect to empirical methods, we can also mention the paper of Singh and Roy (2009), where, based on the examination of 156 published case histories and using the ratio of the peak horizontal ground acceleration and the yield acceleration as the estimator, the authors proposed a correlation relationship for the earthquake-induced deformation of earthen embankments.

In recent years, more sophisticated computer-based linear or non-linear methods for seismic analyses of embankments have been developed, using one-, two- (e.g., Kishida et al., 2009, Athanasopoulos-Zekkos and Seed, 2013) or three-dimensional (e.g., Wang et al., 2013) models. At the same time, Kishida et al. (2009) concluded that simplified models based on equivalent-linear analyses can provide reasonably accurate results up to moderate ground shaking levels, while nonlinear analyses should be used to evaluate dike responses at stronger shaking levels. We therefore employ a simplified approach, since we are concerned with the study on a regional spatial scale in the areas of low to moderate seismicity.

A seismic risk assessment procedure for earthen embankments is presented in the paper of Rosidi (2007), where the dike fragility was expressed as a function of earthquake-induced slope deformations. Considering different strengthening scenarios, Rosidi (2007) estimated levee failure probabilities in dependence on earthquake ground motion return period, however, possible fragility changes due to flood water elevation was not taken into account in that study.

With regards to flood hazard, Apel et al. (2004) developed fragility curves for assessment of the overtopping failure probabilities based on Monte Carlo simulations. Vorogushyn et al. (2009) extended this approach for piping and micro-instability breach mechanisms based on the formulations of Sellmeijer (1989) and Vrouwenvelder & Wubs (1985), respectively.

Recently, Schweckendiek et al. (2014) presented an approach to include field observations in the Bayesian updating of piping failure probabilities of dikes in the Netherlands. Krzhizhanovskaya et al. (2011) reported an integration of reliability



analysis for various breach mechanisms into a prototype flood early warning system, including dike failure and associated inundation modelling. A summary of research and practical methods for reliability assessment of dike systems, considering different failure mechanisms, can be found in Wolff (2008).

The reviewed studies, however, used a single-hazard approach focusing on either earthquake or flood impacts on the
infrastructure. The present study aims at filling the existing methodological gap, considering both hazards together.

The main goals of the study include (1) developing a methodological approach for multi-hazard fragility and damage risk analyses of earthen dikes in earthquake and flood prone areas, and (2) constructing multi-hazard fragility functions for the dikes to be incorporated into the regional flood hazard and risk assessment models and to be used for further risk assessments in the area around Cologne.

The existing regional Inundation Hazard Assessment Model IHAM (Vorogushyn et al., 2010) considers three breach mechanisms: overtopping, piping and micro-instability of the dike slope. More details on the parameterization of these breach mechanisms and the development of respective fragility functions are given in Apel et al. (2004) and Vorogushyn et al. (2009). Here we consider another possible failure mechanism – earthquake-triggered physical damage to earthen dikes due to liquefaction. This type of phenomena may occur in earthquake prone areas, where water-saturated sandy soils have
the potential to liquefy when subjected to seismic vibrations. During liquefaction, when as a consequence of increased pore water pressure the strength of bonds between soil particles is drastically reduced to essentially zero, soil deposits may lose their bearing capacity and behave as fluids (e.g., Kramer, 1996, Idriss and Boulanger, 2008). Other conditions being equal, water-saturation and vibration are major causes of this phenomenon. Therefore, the occurrence probability of liquefaction can predictably increase in multi-hazard (earthquake and flood) prone areas. In our study, we assume that the liquefaction
occurrence in the dike body may result in the subsidence of the dike core as well as in large slope deformations. The subsequent breach of the affected dike section is therefore assumed.

The area under study, along with the communities at risk and location of dikes along the Rhine River, is presented in Fig. 1, where the series of points correspond to the geometric centres of the existing dike sections of about 500-600m length. Also, for the purposes of illustration and general characterization of the area, Fig.1 shows the grid of administrative boundaries
(communities) as well as the general zonation of the seismic hazard. The shown hazard estimates are based on the earlier map of Grünthal et al. (1998), in terms of EMS intensities for an exceedance probability of 10% in 50 years, and are referred to the centres of communities (Tyagunov et al., 2006a). For this study, however, we will calculate more accurate seismic hazard estimates for all the dike locations, as will be shown below.

## 2.    Data and Method

We consider the probability of an earthen dike failure in terms of liquefaction potential, which is estimated with the help of the approach developed by Seed and Idriss (1971). Following this widely-used approach, the liquefaction potential of a site can be characterized by a so-called Factor of Safety (FS) against liquefaction. FS is calculated as the ratio of two values:



first, Cyclic Resistance Ratio (CRR), reflecting the capacity of the soil to resist liquefaction and, second, Cyclic Stress Ratio (CSR), depending on the level of seismic ground shaking.

For estimating the CSR value the following expression is used:

$$CSR = 0.65 \cdot \frac{a_{max}}{g} \cdot \frac{\sigma_{vo}}{\sigma'_{vo}} \cdot r_d \text{ ,} \qquad (1)$$

where $a_{max}$ is the peak horizontal ground acceleration, $g$ is the gravitational acceleration, $\sigma_{vo}$ and $\sigma'_{vo}$ are the total and effective vertical overburden stresses, respectively, and $r_d$ is a stress reduction factor (depending on the depth of the soil layer). Here, for the calculation of the stresses, we take into consideration the water level in the river, which influences the phreatic surface and degree of saturation in the dike core.

For estimating the soil resistance to liquefaction (and, correspondingly, the CRR value) there are different methods exist

(e.g., Youd et al., 2001, Kramer and Mayfield, 2007), in particular, standard penetration testing (SPT), a common in situ testing method for determining the geotechnical engineering properties of soils. However, due to the lack of geotechnical information in the study area, we estimate the penetration resistance approximately using the correlation between the angle of internal friction and SPT-values for sandy soils (Table 1, Peck, 1974).

In addition to the angle of internal friction, for modelling and analysis of the liquefaction resistance of earthen dikes, we also

consider specific weight, porosity and fines content, which are presented in Table 2. The values for the specific weight and friction angle were taken from Vorogushyn et al. (2009) and the references therein. The fines content values are adapted from a dike at the Rhine River in the Netherlands (Van Duinen, 2013). We assume that these soil properties can appropriately characterize the existing earthen dikes along the Rhine.

The performance of the dikes under seismic ground-motion loading is analyzed using a simplified one-dimensional model

assuming that below the water level the soil is in a saturated state. Hence, the phreatic line within the dike body is assumed to be horizontal (obviously, this is a conservative assumption that presumes the sufficiently long duration of the flood water rise). A cross-section of the generic dike model is shown in Fig. 2.

For the development of dike fragility curves, we assume a generic dike height of 5 meters. When integrated into the dynamic flood-earthquake hazard model, the actual dike height and corresponding water level will be taken into account.

In the computational algorithm, the material properties of the dikes are assumed to be homogeneously distributed throughout the cross-section of the dike body; however, they can vary spatially along the river, from one cross-section to another, keeping in mind the range of existing uncertainties of the geotechnical parameters as specified in Table 2.

For quantifying the liquefaction potential, the values of CSR (correlating with the level of seismic hazard) and CRR (depending on the properties of the dikes and the inundation level) are calculated for all points of the dike cross-section from

the crest to the bottom (with a discretization interval of 5 cm). Once both the CSR and CRR values have been determined at a certain point under certain load conditions, we can calculate the factor of safety against liquefaction (FS) employing the relationship (Seed and Idriss, 1971):



$$FS = \frac{CRR}{CSR} \qquad\qquad\qquad (2)$$

At the points where the stress level (CSR) exceeds the resistance (CRR), i.e., the value of FS is below 1, we expect the occurrence of liquefaction, which can cause the development of significant deformations of the earthen structure and, consequently, can lead to the functional failure of the dike.

In this study, we neither analyze the degree of soil deformations caused by liquefaction nor consider the variety of possible damage states of the affected earthen structure. Instead, as a first approximation, we conservatively assume that the initiation of liquefaction (FS ≤ 1) in any point throughout the earthen body of the dike corresponds to the failure (and loss of function) of the protection dike. In other words, the limit state corresponding to the probable breach in the dike section due to earthquake-induced liquefaction is defined as FS = 1.

In view of the uncertainties in the geotechnical parameters of the dikes, calculations of the liquefaction potential (in terms of FS) are implemented using Monte-Carlo simulations (MCS) considering the variability range of the parameters of the dikes presented in Table 2. Based on a frequency analysis of the MCS results, dike failure probabilities are computed for different points of the discretized two-dimensional load space, considering possible combinations of peak ground acceleration and flood water level.

## 3.      Fragility surface

Unlike the commonly used single hazard fragility analysis (when the damage probability is expressed as a function of a single hazard parameter), a multi-hazard fragility analysis should properly take into account all of the relevant hazards and their possible combinations and therefore the fragility relationship should be presented in the corresponding multi-dimensional form. Thus, in the considered case of a dike subjected to two hazards (earthquake and flood), we present the

calculated fragility results in the three-dimensional form, where two horizontal axes represent the space of different possible combinations of the two hazards, while the vertical axis specifies the damage (failure) probability. The developed fragility surface for the earthen dikes is shown in Fig. 3, where the points constituting the surface correspond to the occurrence of the limit state (FS = 1) related to the dike failure due to earthquake-triggered liquefaction. Therefore, the fragility surface defines (on the interval from 0 to 1) the conditional failure probability of earthen dikes as a function of both the seismic (PGA level)

and flood (impoundment level) loads.

Considering the fragility surface as a whole (Fig. 3) one can get a general idea about the main features of the probable dike performance under the multi-hazard conditions, in particular, one can see that, as should be expected, the damage probability for the dikes is proportional to the level of ground shaking, continuously increasing from 0 to 1. At the same time, an increase in the water level can lead to an increased damage probability, even at lower levels of PGA.

To investigate more details and consider additional aspects required for the quantitative fragility analysis of the structures, the fragility surface can be shown as a set of iso-lines corresponding to different percentiles of the calculated distribution of



the FS values, as presented in Fig. 4. The presented iso-lines correspond to the occurrence of the limit state (FS = 1) and specify the failure probabilities in the two-dimensional space of hazards (in units of PGA and flood water level), which are the prerequisites (thresholds) of the initiation of liquefaction in the dike body.

On the left edge of the graph (Fig. 4) one can see that for the water level at the toe of the dikes (without extra-flooding) the

PGA threshold ranges from 0.15 g to 0.54 g for the interval from 1 to 99 percentiles (covering 98% of all calculated values) and from 0.17 g to 0.42 g for 5 to 95 percentiles; the median value marks the level of 0.26 g. When the flood water rises up to about 0.7 - 0.8 m, it has no visible effect on the PGA threshold, while further increases in water levels lead to a considerable shift towards lower PGA values (and this change is practically linear). On the other edge, when the flood water level reaches the top of the structure, the PGA values (and therefore the liquefaction occurrence probabilities) change

significantly. In comparison with the initial state (water level at the toe of the dikes), when the water level equals the crest height, the PGA threshold values decrease to between 0.07 - 0.24 g (for the interval from 1 to 99 percentiles) and to 0.08 - 0.19 g (for 5 to 95 percentiles), while the mean PGA value indicates a level of 0.12 g. Comparing the values for the two edge cases, one can see that, following the water rise, the liquefaction triggering PGA threshold values decrease more than half and concurrently the spread of the values becomes considerably narrower.

The comparative analysis above indicates that a rise of flood water level can lead to an increase in the fragility (and, correspondingly, the damage probability) of the earthen dikes and, therefore, this effect of impoundment should be taken into consideration when analysing the performance of the flood protection earthen dikes in multi-hazard (earthquake and flood) environment. This calls for implementation of a dynamic flood model capable of estimating the impoundment level at the dikes for the period of flood wave propagation in the river system. The developed fragility curves provide thus a basis for

dynamic (unsteady) analysis of flood hazard along the river.

In addition to the three-dimensional fragility surface (Fig.3), displaying the fragility of the structure in the continuous form, Fig. 5 gives an alternate presentation of the calculated results in the form of the discrete fragility functions (more conventional for single hazard analyses), showing the relationship between the damage occurrence probability of the dikes and the level of seismic ground shaking. The set of six fragility functions is presented, each of which includes the influence

of the flood water level, for the six discrete states (from 0 – i.e., at the toe of the dike, to 5 m – i.e., reaching the top of the dike).

As can be concluded, considering the usable range of the liquefaction triggering PGA values (Fig. 5), the developed dike fragility model may find practical application in regions of low to moderate seismicity. For the lower PGA values (0.15 - 0.30 g) the contribution of the effect of impoundment can be more critical than for the higher PGA (when earthquake ground

shaking is sufficiently strong to trigger liquefaction under conditions without extra-flooding).

The presented fragility relationships (which can be used either in the form of the integral fragility surface or as a set of fragility functions for discrete hazard levels) related to the dike damage due to earthquake-triggered liquefaction are essential for the assessment of probability of the failure of earthen flood protection structures in earthquake and flood prone areas, where the effect of interaction between flooding and seismic loading should be considered in risk computations. At the same





time, we note that the presented fragility estimates should be considered as preliminary, bearing in mind, in particular, the simplifications of the one-dimensional dike performance model used in the computations, as well as the conservative assumption about the dike failure even if liquefaction occurs in one point of the dike body. Needless to say, the validation of the models is required as an indispensable consequence of any kind of modelling.

**4.        Dike failure probability assessment**

For dike failure probability assessment, the developed multi-hazard fragility functions should be combined with the probabilistic hazard estimates (including both earthquake and flood) for the area of interest.

As mentioned above, the obtained results are integral to multi-risk analyses in earthquake and flood prone areas around Cologne and aimed to be used for generating a series of flood scenarios with different return periods (from 100 to 1000

years) for the Rhine River reach between Andernach (Rhine-km 613.8) and Düsseldorf (Rhine-km 744.2). Those flood scenarios will take into consideration the probable interaction of the earthquake and flood hazards in the area.

Keeping this purpose in mind, the seismic hazard calculations were implemented for all locations of the earthen dikes on both sides of the Rhine River (as shown in Fig. 1). The input data for the seismic hazard analyses were taken in accordance with the regional model of Grünthal et al. (2010). The hazard calculations were implemented using the GEM (Global

Earthquake Model) OpenQuake software (Crowley et al., 2011a, b) for soil sites characterized by 300 m/s S-wave velocity in the uppermost 30 m, which was assigned considering the results of previous engineering-seismological studies in the area (Tyagunov et al., 2006b, Parolai et al., 2007). The set of calculated seismic hazard curves (in terms of PGA) characterizing the range of probable level of ground shaking for the different dike locations is shown in Fig. 6. In total, 339 dike sections are analysed: 157 of them are on the left side and 182 on the right side of the river.

The calculated PGA values vary in space for different points along the river stretch and the probable level of ground shaking depends on the return period of interest. Thus, for the level of exceedance probability of 10% in 50 years (which is the common standard in the practice of earthquake engineering and corresponds to an average return period of 475 years) the PGA estimates vary over a range of about 0.06 – 0.15 g. For a shorter return period of 100 years, PGA varies in the range of about 0.03 – 0.06 g, whereas for a longer return period of 1000 years the range is about 0.08 – 0.20 g. Note, however, that

for the return periods longer than 1000 years, even higher levels of ground shaking are probable in the area and such low probability phenomena in reality cannot be ruled out.

A remarkable fact is that the spread in the calculated PGA values is not very large, because the course of the Rhine River (and correspondingly the locations of flood protection dikes) closely follows the shape of the seismic hazard zones around Cologne (Grünthal et al., 1998, DIN 4149, 2005), therefore the seismic hazard distribution in the area under study (Fig. 1)

appears rather uniform.

On the basis of the obtained results and referring to the liquefaction susceptibility categorization for different soil types (Youd and Perkins, 1978, HAZUS-MH, 2003), one can make a qualitative conclusion that in this area, there is a risk of dike





damage due to liquefaction induced by seismic ground shaking. It is worth mentioning here that according to observations from past earthquakes (Sasaki et al., 2004) seismic damage to river dikes can be triggered by PGA of 0.16 g or higher. At the same time, according to the observations of Santucci de Magistries et al. (2013) and Quigley et al. (2013), the PGA threshold for liquefaction occurrence can be even less than 0.10 g.

The dike failure probabilities can be quantified by considering the probabilities of occurrence of the earthquake ground shaking level and flood return periods at different dike locations combined with the presented fragility curves. Therefore, the total failure probability can be calculated from the following expression:

$$P(F) = \iint P\big(F\big|S_i, W_j\big) * P(S_i) * P\big(W_j\big)\, dSdW, \qquad\qquad\qquad (3)$$

where $P\big(F\big|S_i, W_j\big)$ is the conditional failure probability given that the combination of the seismic ground shaking $S_i$

and the water level $W_j$ takes place;

$P(S_i)$ is the probability that the seismic input $S$ corresponds to the level *i*;

$P(W_j)$ is the probability that the water level $W$ corresponds to the level *j*.

In other words, the first term of the equation represents the conditional failure probabilities for the dikes due to liquefaction, which can be obtained from the multi-hazard fragility surface (Fig. 3), while the second and the third terms represent

probabilistic estimates of the earthquake (PGA level) and flood hazard (water level) at the dike locations and can be obtained from the corresponding hazard curves.

For the situation without impoundment of the dikes, the dike failure probability due to earthquake-induced liquefaction can be estimated using the simplified form of the equation above, in particular, omitting the influence of the water elevation and considering only one hazard parameter – seismic ground shaking, i.e., in fact, using the traditional single hazard approach.

With the use of the single hazard approach, we find out that it is likely that the earthquake-triggered liquefaction (and therefore dike damage) may occur at some of the considered dike locations even under natural conditions without extra-flooding, though its occurrence probability is not very high. Preliminary quantitative estimates assuming no impoundment of the dikes, show that the level of liquefaction occurrence and, correspondingly, the damage risk for the dikes located in different points along the Rhine River (Fig.1) varies (in dependence on the level of seismic hazard, Fig.6) within the range of

1 - 5*10⁻⁴ per year.

Perhaps, the dike damage risk itself (without taking into consideration effects and consequences of possible floods) may not generate much interest to practitioners. However, one should bear in mind the essential level of existing flood hazard in the area as well as possible temporal coincidence of flooding and strong earthquakes. Actually, the current design criteria of fluvial dikes take into account only flood hazard and do not consider potential multi-hazard impact. Therefore, in case of

probable temporal coincidence of flooding and strong earthquakes, dike protection structures may fail due to liquefaction at





flood return periods below the design level. This may lead to perplexity and negatively affect population, infrastructure, and flood response, requiring emergency actions.

The comprehensive quantitative analysis of the performance of the flood protection system of dikes, including the joint probability of seismic and flood events and their probable interactions in time and space over the whole area, however, is not

a straightforward task and goes beyond the scope of this paper. Here, just for the illustration of the practical application of the developed fragility functions, we present an example of estimating the damage risk for a single dike section.

Exemplarily, for a left-side dike section at Rhine-km 668 near the town Wesseling (south to the city of Cologne, Fig.1), the average maximum water levels were estimated for three return periods 200, 500 and 1000 years, using a dynamic probabilistic-deterministic coupled 1D-2D model (Vorogushyn et al., 2010) setup for the study area at the Rhine River

within the EU-FP7 MATRIX project (Garcia-Aristizabal and Marzocchi, 2013). The hydraulic model uses the flow records at gauge Andernach (Rhine-km 613.8) for estimation of hydrographs and corresponding return periods. Hydrographs are then routed with a coupled 1D-2D model considering dike breaches and associated inundation. The estimated water levels at the selected location are: for the 200-year return period ($p=0.005$ per year) 50.38 m asl (above sea level); for 500-year ($p=0.002$) and 1000-year ($p=0.001$) 50.49 m and 50.52 m asl, correspondingly. The small difference between the calculated

estimates can be explained by the used IHAM model, which considers dike breaches, i.e. the water level at one dike location depends on performance of other dike sections (e.g., if one of the upstream dikes fails, the water outflow would reduce the flood loads on the downstream dike sections). Hence, with increasing return period, the water level it the dike of interest is increasing marginally due to simulated upstream dike breaches.

Assuming the height of the dike of 5 metres at the selected location, the dike would be impounded by 4.50 metres during a

200-year flood event. Correspondingly, the estimated impoundment level would reach 4.61 m for the 500-year and 4.64 m for the 1000-year flood scenarios. These values are used in the illustrative example as the flood hazard input parameters. As for the earthquake hazard at the location, we employ the calculated seismic hazard curve for Wesseling (which belongs to the most upper part of the curves shown in Fig.6).

The estimated probability of liquefaction occurrence in the dike body due to seismic vibration under normal conditions

(without extra flooding), which, in point of fact, reflects the single hazard risk for the selected dike at Wesseling, is about $4.7*10^{-4}$ per year. Considering the combined effect of the two hazards as described in the previous paragraphs, one can see that, on the one hand, their interaction may increase the probability of liquefaction occurrence, though, on the other hand, the probability of the multi-hazard interaction itself decreases proportionally to the product of the single hazard probabilities. Needless to say, different multi-hazard interaction scenarios have different occurrence probabilities and all of them

contribute to the total risk. With the use of the equation (3) we obtain for the 200-year flood scenario the damage probability value about $1.2*10^{-5}$ per year, for the 500-year flood – about $4.9*10^{-6}$ per year and for the 1000-year flood – about $2.5*10^{-6}$ per year. All these values contribute to the total risk value and properly the multi-hazard damage probability for the dike should be integrated over the entire range of flood return periods (probabilities). Consequently, it is reasonable to think that



consideration of the whole range of the multi-hazard interaction scenarios (covering the complete hazard curves) may essentially increase the total (multi-hazard) risk of the dike damage in comparison with the single hazard risk level.

Keeping in mind the adverse cascading effects resulting from the possible dike damage in case of the temporal coincidence of flooding and strong ground shaking, the multi-hazard risk cannot be categorized as negligible for decision making. As the area under study is densely populated and characterized by high concentration of valuable exposure, one should bear in mind that probable failure of the flood defence system may be fraught with far-reaching disaster consequences. This should be the scope of future research.

More detailed and comprehensive quantitative damage risk estimates for the flood protection system and potential consequences for the entire area under study will be produced in a subsequent modelling study considering dynamic hydraulic load and dike breach dependencies along the river course. It will combine the seismic and flood hazard assessments with the newly developed multi-hazard fragility functions for liquefaction as well as with those related to other probable dike breach mechanisms.

## 5.  Conclusions

A simplified methodological framework for multi-hazard fragility and damage risk (failure probability) analyses of fluvial earthen dikes in earthquake and flood prone areas and an exemplary application is presented. The system of earthen dikes along the Rhine River in the area around Cologne is analysed, considering their possible damage due to liquefaction induced by seismic ground shaking in combination with flooding. As a simplifying approximation, we conservatively assume that the liquefaction occurrence at any point throughout the earthen dike body corresponds to the failure (breach) of the dike.

The damage potential of the earthen structures is presented as a fragility surface showing the failure probability as a function of both earthquake ground shaking (PGA) and flood water level (impoundment of the dike). Quantitative fragility analysis shows that a rise in flood water level reduces the liquefaction triggering PGA threshold, leading, therefore, to an increase in fragility and, correspondingly, the failure probability of the dikes. Therefore, this effect should be taken into consideration when analysing the performance of the flood protection earthen dikes in multi-hazard (earthquake and flood) prone areas.

The combined consideration of the obtained fragility estimates and the seismic hazard calculated at the dike locations along the Rhine River allows us to conclude that in the area around Cologne, there is a risk of damage to the earthen dikes due to earthquake-triggered liquefaction even without impoundment of the dikes. In case of temporal coincidence of flooding and strong earthquakes, the risk of damage to the dikes and therefore the consequential impacts can increase. The total risk (failure probability) for the dikes under multi-hazard conditions should be estimated combining the multi-hazard fragility surface with the full-range flood and earthquake hazard curves and taking into consideration different possible combinations of ground shaking and water levels.

We conclude that the level of the damage risk for the flood protection dikes due to earthquake-induced liquefaction cannot be categorized as negligible and should be taken into account in the risk calculations and disaster management policy in the





region under study. Otherwise, the occurrence of dike failures due to a combination of the two hazards may catch stakeholders by surprise and lead to disastrous consequences, since unexpected and unprepared for. The consideration of such multi-hazard scenarios in risk modelling and management would raise awareness and help avoiding the negative surprises at lower costs.

## 5   Acknowledgements

The research leading to these results has received funding from the European Community's Seventh Framework Programme [FP7/2007-2013] under grant agreement n° 265138.

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





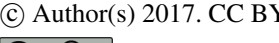

**Figure 1: Location of flood protection dikes along Rhine and the spatial distribution of seismic hazard in the study area. The points correspond to the geometric centres of the existing dike sections on both sides of the river. The hazard estimates are given in terms of EMS intensities for an exceedance probability of 10% in 50 years (Grünthal et al., 1998) and are referred to the centres of communities.**



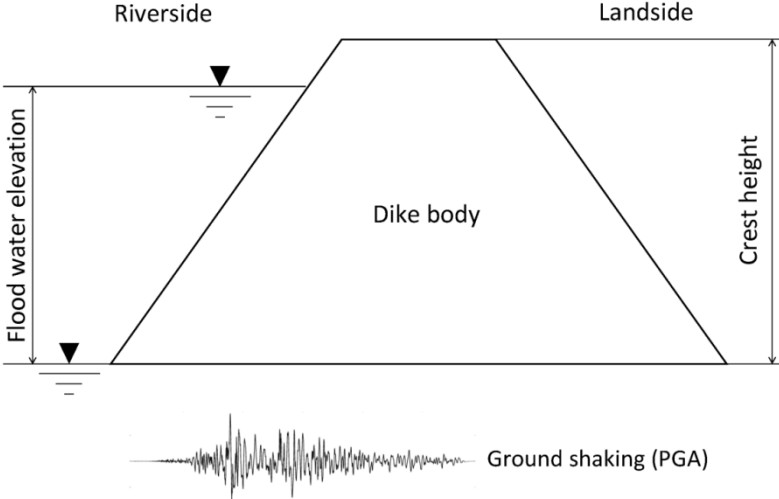

**Figure 2: Generic dike model for earthquake-flood-dike interaction studies**

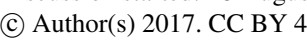



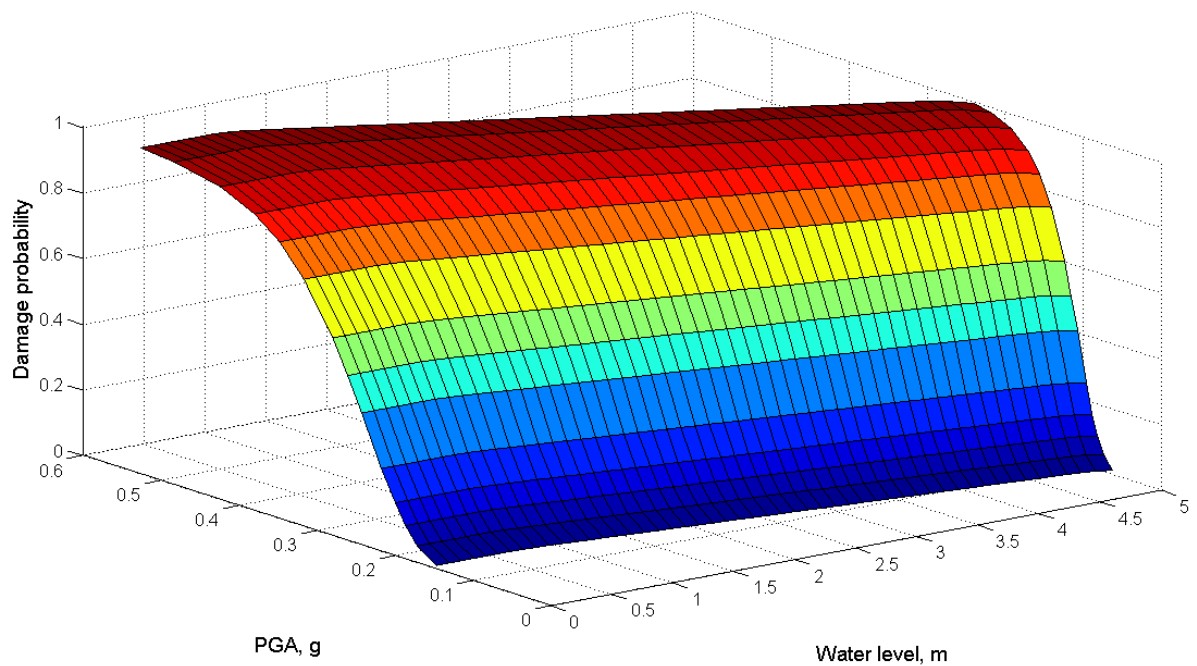

**Figure 3: Multi-hazard fragility surface for the dikes depending on ground shaking level (PGA, g) and water level (m)**




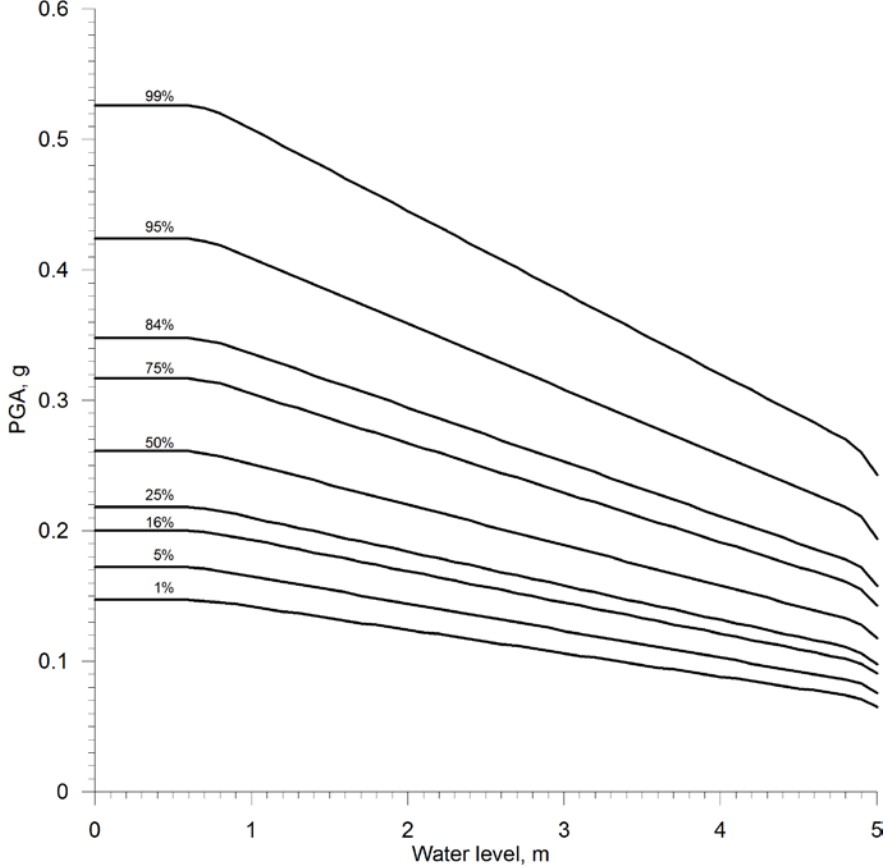

**Figure 4: Dike damage occurrence probability (percentiles) depending on PGA and flood water level**

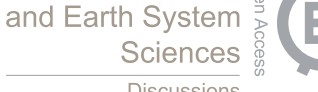
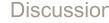


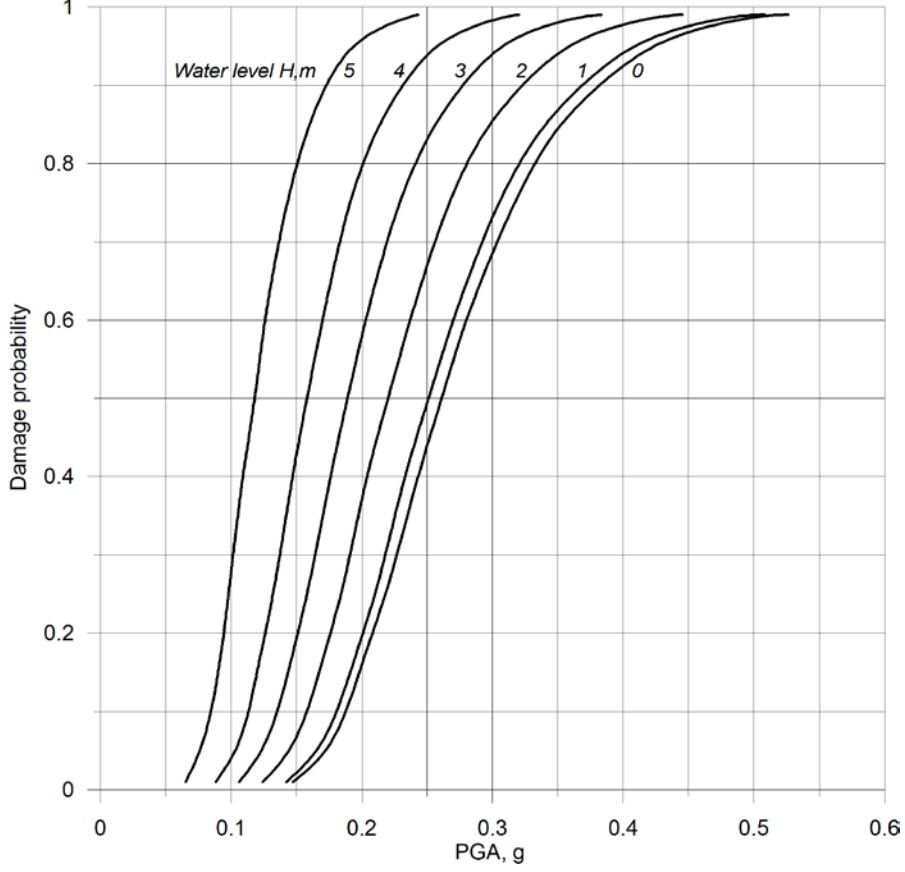

**Figure 5: Fragility functions for the earthen dikes as a function of damage (failure) probability vs. PGA for different water levels (from 0 to 5 m)**

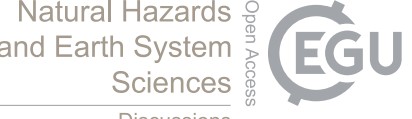



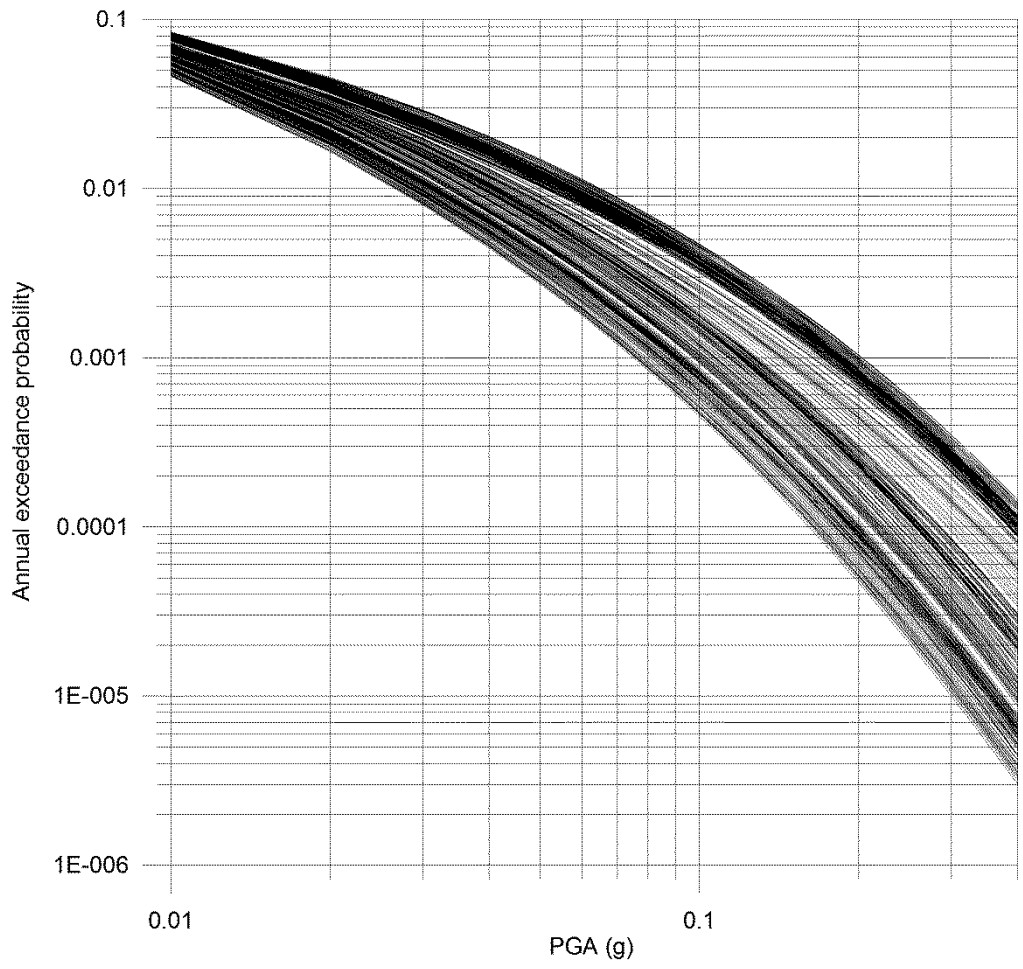

Figure 6: Seismic hazard (mean) curves for the locations of the dikes along the Rhine River (see Fig. 1).



Table 1: Relationship between the angle of internal friction and SPT-values (Peck, 1974)

| SPT, N-Value (blows/ foot) | Density of sand | φ (degrees) |
|---|---|---|
| <4 | Very loose | <29 |
| 4 – 10 | Loose | 29 - 30 |
| 10 – 30 | Medium | 30 - 36 |
| 30 – 50 | Dense | 36 - 41 |
| >50 | Very dense | >41 |



Table 2: Geotechnical parameters of dikes

| Soil properties | Mean | Standard deviation | Minimum | Maximum |
|---|---|---|---|---|
| Specific weight γ (kN/m3) | 18 | 1 | 13 | 21 |
| Friction angle ϕ | 29.2 | 0.3 | 20.8 | 37.6 |
| Fines content FC (%) | 5 | 1 | 3 | 11 |

