# Peer review of "Multi-hazard fragility analysis for fluvial dikes in earthquake and flood prone areas"

_Natural Hazards and Earth System Sciences, 2017_

## Referee Comment (RC1) · Anonymous Referee #1 · 29 Nov 2017

**Review of "Multi-hazard fragility analysis for fluvial earthen dikes in earthquake and flood prone areas" by Tyagunov et al.**

**1    General comments**

The Authors have studied fluvial dikes along the river Rhine nearby Cologne (Germany) under combined seismic and

5    flooding loads. The manuscript contains: multi-hazard fragility analysis and damage risk (failure probability) analysis. It represents an interesting interdisciplinary research, which perfectly fits in with the scope of NHESS. The topic is timely and innovative.

Although the text is generally very well written (it is virtually word-perfect), the style is slightly verbose. There is room for considerably shortening the manuscript by focusing more strictly on the key messages and avoiding redundancy. For

10    instance, the text on P7 (comments of Figs. 3; 4 and 5) is unnecessary long, …

The literature review is comprehensive.

It seems that the Authors focus solely on "liquefaction" of the dike (based on Seed and Idriss, 1971), while worldwide dike overtopping is by far the most frequent mode of dike failure. A discussion is needed in this regard.

Wording "damage risk" sounds a bit odd. If I understand well, the authors use the word "risk" for "probability", since they

15    mean actually the "probability that some damage occurs" (elsewhere, they use "damage probability", e.g. in Sect. 3, in title of Sect. 4 …). In science and engineering, risk is a broader concept than just "probability". It would be wiser to consistently use the wording "failure probability" throughout the manuscript, instead of "damage risk".

Clarifications are necessary regarding the derivation and characteristics of the fragility surface displayed in Fig. 3. Why does the failure probability not reach 1 for the highest values of water level (e.g. overtopping or nearly overtopping conditions)

20    when PGA is low or zero? The same applies for Fig. 4. The reason relates probably to the liquefaction mechanism which is considered by the Authors; but still the results seem a bit puzzling.

**2    Specific comments**

- In the Introduction, mention the different failure mechanisms of dikes (incl. overtopping, seepage …) and briefly discuss their relative importance.

25    - Make clear which are the differences between *embankments* (frontal / normal to the flow direction) and *dikes* (parallel to the flow direction), and which are the consequences in for risk analysis (different designs, presence of a core …)?

- Explain the complementarity between "large scale" studies such as the present one and more detailed small-scale studies (e.g. Rifai et al. 2017, WRR). While the latter are interested in the fine details of the failure mechanisms,

studies such as the present one provide valuable insights on the effects on dike failure at a much broader spatial level (regional).

- Define "hazard curve".

- Is the wording "impoundment of the dike" standard in the field? It sounds a bit odd compared to more standard terminology such as "overtopping" or "overflowing" of the dike …

- Explain shortly "overburden stresses", as the readership of NHESS is multidisciplinary.

- Is "phreatic surface" a standard terminology in English? Does it stand for "water table"?

- Table 1: explain "N-values", "blows/foot".

- Acronym PGA must be clearly defined when it is first used.

- P6 L28: is the word "proportional" (i.e. a purely linear relationship) appropriate?

- Fig. 3 and Fig. 4 seem redundant … They display the same information, don't they?

- P8 L9: why disregard more frequent floods than the 100-year flood?

- P8 L15: explain briefly "S-wave velocity" for he multidisciplinary readership of NHESS.

- P8 L33: is the word "risk" appropriate there?

- P9 L13: replace "term of the equation" by "factor in the integral".

- P9 L17-19: remove this paragraph as it sounds trivial.

- P10 L17: "at" instead of "it"

- P10 L23: "uppermost" instead of "most upper"

- Fig. 6 : the caption must explain that each curve corresponds to a different dike section. Using a grey scale (or colors) for the different curves would make the graph more informative by suggesting which curves correspond to more upstream (resp. downstream) dike sections.

- Conclusion: please shorten. There are some repetitions, particularly in the second half of the Conclusion.

**3    Formal issues, typos …**

- P5 L9: "there are different methods exist" … Rephrase.

- P5 L11: remove "engineering"

- P5 L25: remove "to be"

- P6 L20: "in three-dimensional form", instead of "in the three-dimensional form"

**4 Conclusion**

I strongly recommend that the Authors are invited to submit a revised version of the manuscript for publication in NHESS. I believe that substantially shortening the text, by focusing on the main points, would enhance the potential impact of the paper. If necessary, I am available to review the revised manuscript.

---

## Referee Comment (RC2) · Anonymous Referee #2 · 11 Jan 2018

[referee-annotated manuscript omitted]

---

## Author Comment (AC1) · 19 Feb 2018

**Reply to the comments of the Reviewer #1**

*A: We thank the Reviewer for a comprehensive review of the manuscript, his/her valuable comments and an overall positive evaluation. We respond hereafter to the specific comments of the Reviewer and point out, how we would tackle the raised issues in the revised manuscript.*

R: The Authors have studied fluvial dikes along the river Rhine nearby Cologne (Germany) under combined seismic and flooding loads. The manuscript contains: multi-hazard fragility analysis and damage risk (failure probability) analysis. It represents an interesting interdisciplinary research, which perfectly fits in with the scope of NHESS. The topic is timely and innovative.

Although the text is generally very well written (it is virtually word-perfect), the style is slightly verbose. There is room for considerably shortening the manuscript by focusing more strictly on the key messages and avoiding redundancy. For instance, the text on P7 (comments of Figs. 3; 4 and 5) is unnecessary long, …

*A: We would revise the manuscript in in this regard, shorten the figure captions and remove redundancies.*

The literature review is comprehensive.

It seems that the Authors focus solely on "liquefaction" of the dike (based on Seed and Idriss, 1971), while worldwide dike overtopping is by far the most frequent mode of dike failure. A discussion is needed in this regard.

*A: The reviewer is right that the overtopping seems to be most frequent dike failure mechanisms, as statistics collected by Vorogushyn et al. (2009) shows. However, the methodology for development of fragility curves for overtopping has been already presented by Apel et al. (2004) and to our knowledge is still based on the best available process knowledge considering data availability. Vorogushyn et al. (2009) developed methods for piping and micro-instability failure mechanisms. Though liquefaction is not very common, this is a likely breach mechanism under multi-hazard load by floods and earthquakes. Such fragility curves would thus be required for multi-hazard assessment of dikes in earthquake-prone areas. We shall point out in the manuscript more clearly to the previous developments in the field and the purpose of the presented methodology.*

Wording "damage risk" sounds a bit odd. If I understand well, the authors use the word "risk" for "probability", since they mean actually the "probability that some damage occurs" (elsewhere, they use "damage probability", e.g. in Sect. 3, in title 15 of Sect. 4 …). In science and engineering, risk is a broader concept than just "probability". It would be wiser to consistently use the wording "failure probability" throughout the manuscript, instead of "damage risk".

*A: We agree with the reviewer that the term "damage risk" can be misleading and shall substitute it with the term "damage or failure probability". This is actually what we mean in the presented context.*

Clarifications are necessary regarding the derivation and characteristics of the fragility surface displayed in Fig. 3. Why does the failure probability not reach 1 for the highest values of water level (e.g. overtopping or nearly overtopping conditions) when PGA is low or zero? The same applies for Fig. 4. The reason relates probably to the liquefaction mechanism which is considered by the Authors; but still the results seem a bit puzzling.

*A: Yes, the reviewer is right. The probability of dike failure/damage does not reach 1 even for water levels reaching the dike crest at low values of peak ground acceleration (PGA). At the first glance it looks odd, but if we recall that under probability we mean solely the probability of failure due to liquefaction and not the overall probability of dike failure then this result appears meaningful. We shall briefly explain this in the revised manuscript.*

**2 Specific comments**

In the Introduction, mention the different failure mechanisms of dikes (incl. overtopping, seepage …) and briefly discuss their relative importance.

*A: We shall do so.*

Make clear which are the differences between *embankments* (frontal / normal to the flow direction) and *dikes* (parallel to the flow direction), and which are the consequences in for risk analysis (different designs, presence of a core …)?

*A: It seems that the term 'embankments' is sometimes used as a synonym for dikes/levees. However, "embankment dams" are meant indeed as structures frontal/normal to the flow direction. In the manuscript we refer at some occasions to the literature on "embankment dams". We shall carefully check the use of this term, prove the consistent use and explain the difference as suggested by the reviewer.*

Explain the complementarity between "large scale" studies such as the present one and more detailed small-scale studies (e.g. Rifai et al. 2017, WRR). While the latter are interested in the fine details of the failure mechanisms, studies such as the present one provide valuable insights on the effects on dike failure at a much broader spatial level (regional).

*A: Thanks for this comment. We shall "bridge the scale" by linking small scale process studies to the attempts of applying the knowledge at a larger scale. This is actually what the presented manuscript tries to do: use the detailed geotechnical process knowledge to derive fragility curves which can be used for large scale risk assessment and modelling studies.*

☐Define "hazard curve".

*A: Shall do so.*

☐Is the wording "impoundment of the dike" standard in the field? It sounds a bit odd compared to more standard terminology such as "overtopping" or "overflowing" of the dike …

*A: The literature in the field is not very numerous. So, it is hard to say what is standard though the term was already previously used. "Impoundment of a dam" would be more obvious. With a dike the situation is different, since the flow is usually parallel to the dike. But we still believe, the term "impoundment" would be appropriate since we explicitly do not mean "overtopping" or "overflow" of a dike, but also consider the situations, where a dike is only partially "impounded" by water, i.e. the water level does not reach the crest by far.*

☐Explain shortly "overburden stresses", as the readership of NHESS is multidisciplinary.

☐Is "phreatic surface" a standard terminology in English? Does it stand for "water table"?

☐Table 1: explain "N-values", "blows/foot".

☐Acronym PGA must be clearly defined when it is first used.

☐P6 L28: is the word "proportional" (i.e. a purely linear relationship) appropriate?

*A: We shall address the previous minor comments.*

☐Fig. 3 and Fig. 4 seem redundant … They display the same information, don't they?

*A: In fact, yes. Figure 4 represents the contour plots in the PGA-Water level 2D space of Figure 3. This is because 3D plots are sometimes difficult to interpret, but they nicely show the 3D nature of the fragility surface.*

☐P8 L9: why disregard more frequent floods than the 100-year flood?

*A: Yes, in fact, one can consider also smaller floods as soon as the dikes become impounded. This comment refers, however, to the future modelling study building upon the presented manuscript.*

☐P8 L15: explain briefly "S-wave velocity" for he multidisciplinary readership of NHESS.

*A: Thanks, we shall do so. Surely, for the flood research community, the earthquake-specific terms are not necessarily known.*

☐P8 L33: is the word "risk" appropriate there?

☐P9 L13: replace "term of the equation" by "factor in the integral".

☐P9 L17-19: remove this paragraph as it sounds trivial.

☐P10 L17: "at" instead of "it"

☐P10 L23: "uppermost" instead of "most upper"

*A: The minor issues above will be edited.*

☐Fig. 6 : the caption must explain that each curve corresponds to a different dike section. Using a grey scale (or colors) for the different curves would make the graph more informative by suggesting which curves correspond to more upstream (resp. downstream) dike sections.

*A: thanks, we shall try to implement this suggestion and see if the outcome is better readable.*

☐Conclusion: please shorten. There are some repetitions, particularly in the second half of the Conclusion.
*A: Yes, we shall do so.*

**3 Formal issues, typos …**

*A: We shall properly address the small minor issues listed below*

☐P5 L9: "there are different methods exist" … Rephrase.

☐P5 L11: remove "engineering"

☐P5 L25: remove "to be"

☐P6 L20: "in three-dimensional form", instead of "in the three-dimensional form"

**4 Conclusion**

I strongly recommend that the Authors are invited to submit a revised version of the manuscript for publication in NHESS. I believe that substantially shortening the text, by focusing on the main points, would enhance the potential impact of the paper. If necessary, I am available to review the revised manuscript.

---

## Author Response (AR1)

**Reply to the comments of the Reviewer #1**

*A: We thank the Reviewer for a comprehensive review of the manuscript, his/her valuable comments and an overall positive evaluation. We respond hereafter to the specific comments of the Reviewer and point out, how we would tackle the raised issues in the revised manuscript.*

R: The Authors have studied fluvial dikes along the river Rhine nearby Cologne (Germany) under combined seismic and flooding loads. The manuscript contains: multi-hazard fragility analysis and damage risk (failure probability) analysis. It represents an interesting interdisciplinary research, which perfectly fits in with the scope of NHESS. The topic is timely and innovative.
Although the text is generally very well written (it is virtually word-perfect), the style is slightly verbose. There is room for considerably shortening the manuscript by focusing more strictly on the key messages and avoiding redundancy. For instance, the text on P7 (comments of Figs. 3; 4 and 5) is unnecessary long, …

*A: We have revised the manuscript in this regard, shortened the figure captions and removed redundancies. In particular, we strongly reformulated the text to address the verbose and ornate writing style and make it more concise and pointed.*

The literature review is comprehensive.
It seems that the Authors focus solely on "liquefaction" of the dike (based on Seed and Idriss, 1971), while worldwide dike overtopping is by far the most frequent mode of dike failure. A discussion is needed in this regard.

*A: The reviewer is right that the overtopping seems to be most frequent dike failure mechanisms, as statistics collected by Vorogushyn et al. (2009) shows. However, the methodology for development of fragility curves for overtopping has been already presented by Apel et al. (2004) and to our knowledge is still based on the best available process knowledge considering data availability. Vorogushyn et al. (2009) developed methods for piping and micro-instability failure mechanisms. Though liquefaction is not very common, this is a likely breach mechanism under multi-hazard load by floods and earthquakes. Such fragility curves would thus be required for multi-hazard assessment of dikes in earthquake-prone areas. We have pointed out in the manuscript more clearly to the previous developments in the field and the purpose of the presented methodology. L69-78, L118-126.*

Wording "damage risk" sounds a bit odd. If I understand well, the authors use the word "risk" for "probability", since they mean actually the "probability that some damage occurs" (elsewhere, they use "damage probability", e.g. in Sect. 3, in title 15 of Sect. 4 …). In science and engineering, risk is a broader concept than just "probability". It would be wiser to consistently use the wording "failure probability" throughout the manuscript, instead of "damage risk".

*A: We agree with the reviewer that the term "damage risk" can be misleading and substitutes it with the term "damage or failure probability". This is actually what we mean in the presented context.*

Clarifications are necessary regarding the derivation and characteristics of the fragility surface displayed in Fig. 3. Why does the failure probability not reach 1 for the highest values of water level (e.g. overtopping or nearly overtopping conditions) when PGA is low or zero? The same applies for Fig. 4. The reason relates probably to the liquefaction mechanism which is considered by the Authors; but still the results seem a bit puzzling.

*A: Yes, the reviewer is right. The probability of dike failure/damage does not reach 1 even for water levels reaching the dike crest at low values of peak ground acceleration (PGA). At the first glance it looks odd, but if we recall that under probability we mean solely the probability of failure due to liquefaction and not the overall probability of dike failure then this result appears meaningful. We shall briefly explain this in the revised manuscript.*

**2 Specific comments**
In the Introduction, mention the different failure mechanisms of dikes (incl. overtopping, seepage …) and briefly discuss their relative importance.

*A: This has been shortly discussed, L69ff*

Make clear which are the differences between *embankments* (frontal / normal to the flow direction) and *dikes* (parallel to the flow direction), and which are the consequences in for risk analysis (different designs, presence of a core …)?

*A: It seems that the term 'embankments' is sometimes used as a synonym for dikes/levees. However, "embankment dams" are meant indeed as structures frontal/normal to the flow direction. In the manuscript we refer at some occasions to the literature on "embankment dams". We checked the use of this term. We now consistently use the term "dikes" for the structures parallel to the flow direction and "embankment dams" for structures normal to the flow direction.*

Explain the complementarity between "large scale" studies such as the present one and more detailed small-scale studies (e.g. Rifai et al. 2017, WRR). While the latter are interested in the fine details of the failure mechanisms, studies such as the present one provide valuable insights on the effects on dike failure at a much broader spatial level (regional).

*A: Thanks for this comment. We "bridge the scale" by linking small scale process studies to the attempts of applying the knowledge at a larger scale. This is actually what the presented manuscript tries to do: use the detailed geotechnical process knowledge to derive fragility curves which can be used for large scale risk assessment and modelling studies.*

☐Define "hazard curve".

*A: Done*

☐Is the wording "impoundment of the dike" standard in the field? It sounds a bit odd compared to more standard terminology such as "overtopping" or "overflowing" of the dike …

*A: The literature in the field is not very numerous. So, it is hard to say what is standard though the term was already previously used. "Impoundment of a dam" would be more obvious. With a dike the situation is different, since the flow is usually parallel to the dike. But we still believe, the term "impoundment" would be appropriate since we explicitly do not mean "overtopping" or "overflow" of a dike, but also consider the situations, where a dike is only partially "impounded" by water, i.e. the water level does not reach the crest by far.*

☐Explain shortly "overburden stresses", as the readership of NHESS is multidisciplinary.

A: This has been reformulated

☐Is "phreatic surface" a standard terminology in English? Does it stand for "water table"?

*A: "phreatic surface" is often used in this content. Water table is typically meant to be horizontal, whereas "phreatic surface" can be inclined and develops gradually in the dike core*

☐Table 1: explain "N-values", "blows/foot".

*A: This is a standard number associated with the standard penetration test (SPT) and is explained in any textbook on soil mechanics. The explanation here would be too lengthy.*

☐Acronym PGA must be clearly defined when it is first used.

*A: Done*

☐P6 L28: is the word "proportional" (i.e. a purely linear relationship) appropriate?

*A: This sentence has been reformulated*

☐Fig. 3 and Fig. 4 seem redundant … They display the same information, don't they?

*A: In fact, yes. Figure 4 represents the contour plots in the PGA-Water level 2D space of Figure 3. This is because 3D plots are sometimes difficult to interpret, but they nicely show the 3D nature of the fragility surface. We thus prefer to keep both and shortly discuss the features of both plots.*

☐P8 L9: why disregard more frequent floods than the 100-year flood?

*A: Yes, in fact, one can consider also smaller floods as soon as the dikes become impounded. This comment refers, however, to the future modelling study building upon the presented manuscript. This is made clear in the revised manuscript. This is a valuable comment by the Reviewer. We shortly discuss the implication of multi-hazard analysis for scenarios with flood return period below design level and expected impact on flood risk curves, L283-294.*

☐P8 L15: explain briefly "S-wave velocity" for he multidisciplinary readership of NHESS.

*A: Thanks. "S-wave velocity" is now explained, L300ff*

☐P8 L33: is the word "risk" appropriate there?

*A: This has been revised.*

☐P9 L13: replace "term of the equation" by "factor in the integral".

*A: Done*

☐P9 L17-19: remove this paragraph as it sounds trivial.

*A: revised*

☐P10 L17: "at" instead of "it"

☐P10 L23: "uppermost" instead of "most upper"

*A: The minor issues above are revised.*

☐Fig. 6 : the caption must explain that each curve corresponds to a different dike section. Using a grey scale (or colors) for the different curves would make the graph more informative by suggesting which curves correspond to more upstream (resp. downstream) dike sections.

*A: Thanks, the explanation is added. The distribution of earthquake hazard is indicated in Figure 1. The different colors for curves would not enhance readability.*

☐Conclusion: please shorten. There are some repetitions, particularly in the second half of the Conclusion.
*A: Yes, conclusions have been made more concise.*

**3 Formal issues, typos …**

*A: We addressed the minor issues listed below*

☐P5 L9: "there are different methods exist" … Rephrase.

☐P5 L11: remove "engineering"

☐P5 L25: remove "to be"

☐P6 L20: "in three-dimensional form", instead of "in the three-dimensional form"

**4 Conclusion**

I strongly recommend that the Authors are invited to submit a revised version of the manuscript for publication in NHESS. I believe that substantially shortening the text, by focusing on the main points, would enhance the potential impact of the paper. If necessary, I am available to review the revised manuscript.

**Reply to the comments of the Reviewer #2**

*A: We thank the Reviewer for a positive review of the manuscript, his/her valuable comments. We respond hereafter to the specific comments of the Reviewer and point out, how we tackled the raised issues in the revised manuscript.*

General comments:

[*] I suggest you include a model uncertainty factor in the MCS

*A: This is not quite clear to us what sort of the model uncertainty factor is meant here to be used in the Monte Carlo simulation. So far, we have considered the uncertainty in the geometrical and geotechnical dike parameters by taking into account their moments and typical probability distributions available in the literature. Considering the model (structure) uncertainty requires alternative model formulations i.e. different equations, which are not available in our case. Thus, we do not see, how we can consider model structure uncertainty unless the reviewer means something different under model uncertainty factor.*

[*] I think you need to consider more frequently occurring water levels, not just the ones with very high return periods. The more frequent occurring water levels have higher likelihood of occurring in combination with an earthquake event.

*A: Thanks for this comment, which goes in the same direction with the Reviewer #1. Indeed, smaller flood events are more likely, thus the probability of the coincidence with earthquakes would be higher. In terms of risk (probability x damage), the damage from small floods is however smaller. In any case, this is a valid comment, but the probabilities of floods/flood scenarios will be considered in a subsequent analysis, where we plan to integrate the entire flood and earthquake risk in a Monte Carlo analysis. In this subsequent study, the here developed fragility curves will be used for assessment of dike failures and subsequent inundation using hydrodynamic modelling. This goes, however, beyond the scope of the presented study. We shall consider also scenarios with smaller return periods than 100. We shortly discuss the implication of multi-hazard analysis for scenarios with flood return period below design level and expected impact on flood risk curves, L283-294.*

[*] Please write out in more detail how you get to failure probabilities. I get the feeling you multiply annual flood probabilities (T=200 means p=0.005) with annual PGA probabilities. This is not allowed, which can be seen from the fact that the product has the unit year^-2, which has no meaning.

If that is the case, the method is incorrect. You need to take into account that if in year Y both a flood event and an earthquake event occur, it is more likely that they occur at different times in the year than that they happen at the same time. This needs to be taken into account in the computation. This strongly decreases the failure probability.

Additionally, you need to take into account the recovery (repair) time of the dike after an earthquake. This increases the failure probability.

It appears you did not take these factors into account. Apologies if you have, but in that case I propose you elaborate more on this

[*] I feel this paper should at least do one complete risk computation. Suggesting that it is "reasobale to think that the combination leads to higher risks", as you do near the end of the paper, is not doing the rest of the paper justice. And it should not be that much extra worrk.

Furthermore, your whole introduction is about how important it is to consider the combination of the two hazards (which I agree with). Then the least I expect is a comparison of failure probabilities of [a] a flood risk analysis, [b] an earthquake risk analysis and [c] a combined risk analysis

*A: We thank the reviewer for pointing to these two issues. We would prefer to address them together as they partly relate to each other. In fact, we envisage a subsequent study doing a full-scale multi-risk assessment of "simultaneous" occurrence of floods and earthquakes by running a coupled 1D-2D hydrodynamic model for the Rhine and considering dike breaches (by the way not only due to liquefaction, but also due to overtopping and piping). As partly proposed by the reviewer we intend to compare the marginal change in flood risk due to multi-hazard load in relation to single-flood risk curve. This is, however, much more work contrary to the expectation of the Reviewer since the hydrodynamic and dike breach simulations are complex, run also probabilistically in Monte Carlo simulation and the variety of results need to be evaluated from various perspectives (see e.g. Vorogushyn et al., 2010). We therefore abstain from merging this research with the here proposed methodological development on the derivation of fragility curves for liquefaction in one manuscript. In the analysis we consider the "simultaneous" occurrence of floods and earthquakes if the latter occurs within 30 days period – a typical duration of a flood wave on the Rhine. For the subsequent analysis, we have developed synthetic flood hydrographs of 30 days duration. We shall modify the equation (3) to make this point clear. Yes, the Reviewer is right that multiplying the annual probabilities of earthquake and floods is wrong. Indeed, we mistakenly used annual earthquake probability in combination with annual flood probabilities. We recomputed the failure probabilities for an exemplary dike section using earthquake probability within 30 days window. Assuming the time window of 30 days, we undertake several assumptions. First, the probability of liquefaction depends on the development of the water table within a dike during the onset of the flood event. We treat this probability as uniform for the sake of brevity. Otherwise, we would need to carry out the dynamic modelling of water front propagation, which is an additional serious complication. Second, during the flood event no dike repair actions are taken into account, which might reduce the overall flood risk. This effect is however very difficult to estimate. The assumption of "no repair" during an entire year would be very unrealistic as mentioned by the reviewer. Such an assumption for the 30 days period might be reasonable, but in any case, this represents the conservative risk assessment. These limitations are discussed in the revised manuscript.*

*Minor and editorial remarks: the Reviewer #2 proposed several editing changes in the text using the change track mode in the pdf file. We shall carefully address them all in the revised manuscript, but we do not summarize them here in this reply letter.*

**References:**

[revised manuscript text omitted]
\big(F\big|{\sim}{\sim}S_i^{30}{}_{\sim i}, W_j\big) * P\big(S_i^{30}{\sim}{S}_{\sim i}\big) * P\big(W_j\big)\, dS dW, \qquad (3)$$

where $P\big(F\big|S_i^{30}, W_j\big)$ is the conditional failure probability given  the
combination of the seismic ground shaking $S_i^{30}$ within a time window of 30 days and
the water level $W_j$ ;

$P\big(S_i^{30}\big)$ is the probability of occurrence of  the seismic input $S$
of the level $i$ within a time window of 30 days;

$P\big(W_j\big)$ is the probability that the water level $W$  the level $j$.

The first factor in the integral  represents the
conditional failure probabilities , which can be
obtained from the multi-hazard fragility surface (Fig. 3), while the second and third ones represent probabilistic estimates of the seismic (PGA level) and
flood hazard (water level) at the dike locations and can be obtained from the
corresponding hazard curves.

For the situation without  flooding by combining the seismic hazard curves (Fig.
6) with the fragility curve corresponding to the water level of 0 m (Fig. 5),
the earthquake-triggered liquefaction
may occur at some of the considered dike locations
though the probability is not
very high.
The probability varies in this case
within the range of 1 - 4$\underline{5}$*10$^{-5\underline{4}}$ per year.

The current design criteria of fluvial dikes take into account only flood
hazard and do not consider potential multi-hazard impact. Therefore, in case of
probable temporal coincidence of flooding and strong earthquakes, dike protection
structures may fail due to liquefaction at flood return periods below the design level.
This may lead to perplexity and negatively affect population, infrastructure, and flood
response, requiring emergency actions.

A comprehensive quantitative risk analysis
considering the joint probability of seismic and
flood events and their  interactions in time and space
requires
continuous hydraulic model and multi-hazard integration. This goes beyond the
scope of presented research. Here,  for the illustration
of the developed fragility functions, we present an example for
estimating of the failure probability for a  specific dike section.

For a left-side dike section at Rhine-km 668 near the town Wesseling
(south to the city of Cologne, Fig.1), the average maximum water levels were
estimated for three return periods 200, 500 and 1000 years, using a dynamic
probabilistic-deterministic coupled 1D-2D model (Vorogushyn et al., 2010) setup for the study area at the Rhine River within the EU-FP7 MATRIX project (Garcia-
Aristizabal and Marzocchi, 2013). The hydraulic model uses the flow records at
gauge Andernach (Rhine-km 613.8) for estimation of hydrographs and
corresponding return periods. Hydrographs are then routed with a coupled 1D-2D
model considering dike breaches and associated inundation. The estimated water
levels at the selected location are: for the 200-year return period (p=0.005 per year)
50.38 m asl (above sea level); for 500-year (p=0.002) and 1000-year (p=0.001)
50.49 m and 50.52 m asl, correspondingly.

Assuming the height of the dike of 5 metres at the selected location, the dike would
be impounded by 4.50 metres during a 200-year flood event. Correspondingly, the
estimated impoundment level would reach 4.61 m for the 500-year and 4.64 m for
the 1000-year flood scenarios. The small difference between the calculated
estimates can be explained, in particular, by the used model, which considers
dike breaches upstream, i.e. the water level at one dike location
depends on performance of other dike sections (e.g., if one of the upstream dikes
fails, the water  outflow would reduce the flood loads on the other dike
sections).

Combining the flood hazard estimates with seismic hazard curves and fragility
function for the point of interest, the probability of liquefaction at Wesseling without
flooding is
about
4.7*10$^{-4}$ per year.
Applying
Eq. 3, we obtain for the 200-year flood scenario the
liquefaction failure probability of 1.*10$^{-6}$ per year, for the 500-year flood –

about 4.1\*10$^{-\sout{6}7}$ per year and for the 1000-year flood – about 2.5\*10$^{-\sout{6}7}$ per year. All these  return period scenarios contribute to the total risk value . Consequently, it is expected that the multi-hazard interaction scenarios  essentially increase the total risk level in comparison with the estimated single hazard risk level though the combined probabilities of earthquake and floods are very small.

~~Therefore, as could be expected, in the event of the probable temporal coincidence of flooding and strong ground shaking, the total risk of the dike damage due to liquefaction is increasing. One should bear in mind, however, that, as indicated above, the obtained quantitative risk estimates are calculated solely for the purpose of illustration of the approach and not intended for practical applications.~~

Nevertheless, dike failures due to liquefaction in case of a multi-hazard impact bears the potential of surprise and malign consequences, which should be considered in a comprehensive risk assessment (Merz et al., 2015). In particular, under hydraulic load below the (hydraulic) design level (< 200-year return period at the German Rhine reach), dikes might be considered predominantly safe in a single-type hazard analysis, whereas the occurrence of liquefaction would dramatically change flood inundation patterns and loss distribution. Though not necessarily extreme, but still significantly strong floods and 'unexpected' dike failures in combination may still harmfully affect the densely populated areas with high asset concentration such as floodplains along the Rhine. Hence, a quantitative multi-risk analysis is advocated in earthquake and flood prone areas considering the effect of dike liquefaction despite a relatively small probability of the joint occurrence of both perils. ~~At the same time, based on the obtained results, we may conclude that the level of the failure risk for the 
[revised manuscript text omitted]

---

## Author Response (AR2)

**Author's response to the Editor's comments**

We thank the Editor for supervising the revision round and providing very useful comments. We thank also both Reviewers for their efforts in reviewing the last version of the manuscript.

We addressed the technical comment of the second Reviewer by including the suggested reference.

We agree with the Editor's suggestions and revise the introductory and the results part making the scope of the manuscript clearer. We underscore that the presented study (1) develops a methodology for fragility analysis of dikes due to liquefaction under seismic and hydraulic load, (2) presents a framework for dike failure probability calculation for joint occurrence of earthquake and flood accounting for their respective probabilities. The latter is demonstrated for an exemplary dike section at the middle Rhine River with the seismic load computed for the study area. We point out that the full-fledged flood and seismic multi-risk assessment goes beyond the scope of the presented study. The developed fragility curves form a basis for the subsequent multi-risk analysis, which would involve hydraulic modelling of the channel flow, dike breaching, flood inundation simulation and damage assessment.

We hope the provided modifications highlighted in the 'change-mode' make these points transparent. We believe that the presented methodology for liquefaction fragility calculation and failure probability qualifies as a self-consistent manuscript.

With kind regards,

Sergiy Vorogushyn on behalf of the co-authors

[revised manuscript text omitted]